# Effects of *Urtica cannabina* to *Leymus chinensis* Ratios on Ruminal Microorganisms and Fiber Degradation In Vitro

**DOI:** 10.3390/ani10020335

**Published:** 2020-02-20

**Authors:** Zhenbin Zhang, Shan Wang, Mengzhi Wang, Khuram Shahzad, Xiaoqing Zhang, Ruxin Qi, Liangfeng Shi

**Affiliations:** 1College of Animal Science and Technology, Yangzhou University, Yangzhou, Jiangsu 225009, China; 18762304859@163.com (Z.Z.); sunnyshan5233@163.com (S.W.); ruxinqi@126.com (R.Q.); slf1184270717@163.com (L.S.); 2Department of Biosciences, COMSATS University Islamabad, Park Road, Islamabad 45550, Pakistan; khuramsb@gmail.com; 3Institute of Grassland Science, Chinese Academy of Agricultural Sciences, Huhehote 010010, Inner Mongolia, China; ZhangXiaoqing@caas.cn

**Keywords:** *Urtica cannabina*, *Leymus chinensis*, digestibility, fiber microstructure, microorganisms

## Abstract

**Simple Summary:**

*Urtica cannabina* is an unconventional forage crop that has a wide distribution in northern parts of China and is rich in nutritional values. It has a considerable practical significance in cases of shortage of quality forage, such as *Leymus chinensis*, which is used for the growth and production of ruminants. Our study found that proper addition ratios of *U. cannabina* and *L. chinensis* can increase the fiber digestibility by modulating the rumen microbial community in vitro. We also suggest that proper addition of *U. cannabina* and *L. chinensis* ratios may be one of the nutritional strategies that can improve the digestion and absorption performance of fiber feedstuff and potentially alleviate the shortage of quality forage for ruminants.

**Abstract:**

The study was conducted in vitro to investigate the effects of different ratios of *Urtica cannabina* and *Leymus chinensis* on fiber microstructure and digestibility in ruminal fluid. The experiment was divided into five groups based on the *U. cannabina/L. chinensis* ratios: A (0:100), B (30:70), C (50:50), D (70:30), and E (100:0). The culture medium was collected at 0, 1, 3, 6, 12, and 24 h. The results showed that: (1) in vitro crude protein degradability (IVCPD) was higher in group A, whereas in vitro neutral detergent fiber degradability (IVNDFD) was higher in group C (*p* < 0.05); (2) protozoa count was increased from 1 h to 3 h and decreased afterwards, with significant differences observed in several genera (*p* < 0.05); (3) microbial crude protein (MCP) contents at 1, 3, 6, and 24 h were higher in groups A and C (*p* < 0.05); (4) the basic tissue of *U. cannabina* was gradually degraded. At 24h, the secondary xylem vessel structure was observed in groups B and C, but not in groups D and E. In summary, there was higher neutral detergent fiber (NDF) digestibility, higher rumen MCP contents, and lower protozoa count, showing the significance of the 50:50 ratio for microbial growth and fiber digestibility.

## 1. Introduction

High-quality roughage is of great importance in ruminant production [1]. Currently, the amount of high-quality roughage is inadequate for the ruminant industry in China [2]. *Leymus chinensis*, also known as fake wheat grass or Chinese ryegrass, is a major specie in the grasslands of Inner Mongolia Autonomous Region in the northern part of China, covering a large area of the temperate grasslands of eastern Eurasia [3]. *L. chinensis* is a perennial herb, which is one of the favorite forages of large herbivores because of its high palatability and proper nutritional values [4]. However, long-term livestock farming, dominated by sheep and goats, causes severe degradation by reducing canopy cover significantly [5]. Therefore, alternative forage sources are needed as a substitute for *L. chinensis* to reduce grassland degradation dominated by *L. chinensis* and to alleviate vacancies in high-quality forage. 

To our knowledge, an unconventional forage with a high nutritive values can be used as alternative feed source for ruminant animals [6,7]. High-quality forages rich in bioactive ingredients, such as saponins, which are widely distributed in alfalfa and clover, have been shown to improve the gastrointestinal development and nutrient metabolism of ruminants [8,9]. Concretely, *Urtica cannabina* L., which is widely distributed herbaceous group of stipules in China, is an unconventional forage. Its stems and leaves are rich in vitamins, trace elements, and cellulose, as well as a variety of active substances. The trophic protein content is as high as 35% [10]. To make full use of *U. cannabina* as animal feed, it should have considerable practical significance due to its high content of protein and abundance of active substances. However, the utilization rate of *U. cannabina* is still low, and needs greater research focus to utilize this kind of unconventional forage to solve the shortage of high-quality feed resources in ruminant industry. 

In this decade, the usage of *U. cannabina* has been elucidated in livestock practices. Humphries and Reynolds [11] reported that production levels in terms of milk were maintained when *Urtica dioica* haylage (100 g/kg of dry matter) replaced ryegrass silage in the diet of lactating dairy cows, in addition to a numerical reduction of feed intake. *U. cannabina* has also been fed to animals for centuries in Inner Mongolia, China [12]. A recent study by Jin et al. [13] demonstrated that *U. cannabina* (500 or 250 g/kg in diet) improved the nutrient digestibility and gastrointestinal morphological characteristics of growing lambs. Obviously, rumen microorganisms, such as bacteria, protozoa, fungi, and archaea, perform critical roles in terms of rumen fermentation and nutrient digestibility, especially in fiber degradation [14]. *U. cannabina* might increase nutrient digestibility by modulating the rumen microbial community, as it contains a variety of biologically active compounds, such as histamine, serotonin, and other phenolic compounds and flavonoids, which are currently being used for anti-inflammatory treatment of arthritis [15,16,17]. Kleim et al. [18] found that *U. cannabina* (100 mg/g) in vitro can effectively prevent rumen acidosis by increasing the pH value of rumen fluid to 30%. On the other hand, the active factors in *U. cannabina* might be harmful to rumen microorganisms if an improper ratio of *U. cannabina* is used in animal diets, which influence fiber digestibility [19]. However, the optimal ratio of *U. cannabina* used in ruminant diets has not been very clear until now. For this purpose, different ratios of *U. cannabina* and *L. chinensis* were set as substrates in the current experiment in order to investigate their effects on the fiber microstructure and degradation by rumen microorganisms in vitro, and to establish the optimal ratio of *U. cannabina* and *L. chinensis* for further feeding experiments.

## 2. Materials and Methods

This project was approved on ethical grounds by the Animal Welfare Committee of Yangzhou Veterinarians of the Agriculture Ministry of China (Yangzhou, China, No. 201406018). 

### 2.1. Animal Care and Management

Three healthy Xuhuai goats with permanent rumen fistulas and body weight (BW ± SD) of 35 ± 3.1 kg were selected to provide rumen liquid for in vitro culture from the experimental farms of Yangzhou University. Goats were individually housed in a covered pen (1.7 × 2.8 m) with slatted wood floors and equipped with an automatic drinking bowl and feed chute. The composition of the diet included 70% high-quality alfalfa, 28% corn grains, and 2% soybean meal. It was mixed well and then fed to the animals in the morning at 07:00 and in the evening at 19:00. The amount of dry matter of the feed offered to the animals was about 3% of BW. The animals were provided with ad libitum access to fresh drinking water and mineral licking blocks. 

### 2.2. Plant Material Preparation and Experimental Design

The *U. cannabina* samples in the flowering stage were collected early in the month of July 2017 from Xilinguole, Inner Mongolia region of China. The forage was mechanically collected and dried outside in the field. 

*L. chinensis* was bought from Caodu Co., Ltd., from Xilinhaote, Inner Mongolia region, China, cut and collected automatically by machine, and then dried in an open environment. 

*U. cannbinau* and *L. chinensis* were ground with a Retsch ZM 100 Wiley mill (Retsch GmbH, Haan, Germany) and passed through 1 mm screen after drying at 65 °C in a forced-air oven to a constant mass. This refined mass was used as the substrates for the in vitro culture with the following five ratios 0:100 (group A), 30:70 (group B), 50:50 (group C), 70:30 (group D), and 100:0 (group E). It was rationally mixed as a fermentation substrate (each group of culture substrate is a combination of two kinds of grasses). The ingredients and chemical composition of the substrates are reported in Table 1.

### 2.3. In Vitro Fermentation and Sample Collection

In vitro culture was performed according to the method of Menke and Steingass [20]. The rumen fluid was collected from the rumen fistula using a self-made vacuum negative pressure device before morning feeding (at 07:00). The fluid was filtered through four layers of gauze into a thermos flask that had been previously charged with CO_2_ and preheated at 39 °C. Artificial saliva was made by mixing distilled water, mineral element solution, trace element solution, and resazurin solution. The mixed solution was placed in a triangular flask, heated at 39°C in a thermostatic water bath shaker (SHA-A, Hengfeng Instrument, Jintan, China), and then reducing solution was added while shaking at 39 °C. The CO_2_ was continuously introduced by inserting the air tube connected to the CO_2_ steel bottle (GB/T 23938, purity ≥ 99.9%, China) containing the solution. The bottle was sealed with a sealing film until the color of the solution changed from blue to pink, and finally to a colorless and transparent liquid. The CO_2_ was injected until the pH of the solution was set to 7.0–7.3. The rumen fluid and artificial saliva were mixed at a ratio of 1:2, and CO_2_ was continuously passed until the saturation point was reached. The mixed solution was further used as a culture medium for in vitro fermentation.

Ninety-one fermentation glass bottles (150 mL) were used, which were assigned into two runs. The first run used 75 fermentation glass bottles (5 treatments × 5 incubation times × 3 replicates each time) for the determination of in vitro digestibility, protozoa count, and microbial crude protein (MCP) contents. The second run used 16 fermentation glass bottles (4 treatments × 4 incubation times) to observe the fiber structure of *U. cannabina*. During the first run, the substrate of each group (A-E) was accurately weighed 0.5000 g (±0.0500 g) in the 150 mL fermentation bottles. Then, 75 mL of culture solution was added, and the CO_2_ was introduced through a tube vent at the same time. The fermentation bottle was continuously filled with CO_2_ for about 15 s, and then quickly closed with a rubber stopper. The whole media was cultured in a water bath shaker at 39 °C. In the second run, each group (B–E) was accurately weighed to 0.5000 g (±0.0500 g) according to the ratios, and then *U. cannabina* samples (dried at 65 °C) covered with a nylon bag (pore size: 20 μm) were placed along with nylon bags of *U. cannabina* and the *L. chinensis* samples into the fermentation bottles, and finally cultivated according to the first run.

After the first run, the culture medium was collected into 5 mL centrifuge tubes at 1, 3, 6, 12, and 24 h to determine the MCP contents and the count of the total protozoa population. Finally, the contents of the fermentation flask cultured for 24 h were filtered through a nylon bag (pore size: 20 μm), which was dried at 65 °C to a constant weight in advance (known quality). After filtering, the contents were dried at 65 °C for 24 h to a constant weight to determine the digestibility of dry matter (DM), crude protein (CP), neutral detergent fiber (NDF), and acid detergent fiber (ADF).

After the second run, the nylon bags containing *U. cannabina* samples in the fermentation flask were taken out at 1, 3, 6, and 24 h. Then, these bags were washed with 0.9% saline, opened after being squeezed slightly, then a small amount of sample was collected with a sterile tweezer into a 5 mL centrifuge tube and fixed in 2.5% glutaraldehyde in phosphate buffer (pH = 7) for at least 12 h at 4 °C. The samples were taken for scanning electron microscopy observation to investigate the fiber structure of *U. cannabina*. 

### 2.4. Laboratory Analyses

#### 2.4.1. Digestibility

Dry matter (DM) was analyzed according to the Association of Official Analytical Chemists (AOAC, 2005: ID 973.18) [21] and the crude protein (CP) was analyzed using a Kjeldahl analyzer (Kjeltec 2300; FOSS Analytical AB, Hoganas, Sweden). Neutral detergent fiber (NDF) and acid detergent fiber (ADF) were determined using the filter bag technique (ANKOM 2000; Ankom Technology Corp., Fairport, NY), as described by Van Soest et al. [22], with amylase pre-treatment and by adding the sodium sulfite to the neutral detergent.

Substrate nutrient degradation rate was calculated using the following formula:A = (A_1_-A_0_)/A_1_×100(1)
where A is the substrate nutrient degradation rate (%), A_1_ is the nutrient content in the substrate, and A_0_ is the nutrient content in the residue.

#### 2.4.2. Protozoa Counting 

Protozoa enumeration was performed on a blood cell counting plate. The culture solution sample was filtered through 4 layers of gauze and diluted with methyl green staining solution (containing 8 g/L NaCL, 0.6 g/L methyl green, and 100 mL/L formaldehyde) in equal amounts (1 mL). The protozoa number was counted under an optical microscope (Olympus, Tokyo, Japan) at 100× magnification and classified at 400× magnification power. A pipette with a hole diameter greater than 1.0 mm was used to fill the counting chamber with a continuous flow of sample dilution fluid, then left to stand for 2 min for microscopic examination. The same sample was counted 4 times and the mean value was used for subsequent analysis. The characteristics of the protozoa include a large nucleus, small nucleus, cilia, and bone plate classification count [23]. The genus of protozoa counts include *Entodinium*, *Diplodinium*, *Isotricha*, and others. The number of protozoa was calculated using the following formula [24]:P= N × D × 4 × 10^4^(2)
where P is the protozoa number/mL, N is the total number of protozoa in four middle panes, and D is the dilution factor.

#### 2.4.3. MCP Content

Trichloroacetic acid (TCA) precipitation method was used to detect the MCP content produced by rumen fermentation according to Hall and Herejk [25].

#### 2.4.4. Sample Preparation and Scanning Electron Microscopy

First, the *U. cannabina* samples were washed 3 times with phosphate buffer, then dehydrated in ascending concentration of absolute ethanol (50%, 70%, 80%, 90%, 100%), followed by tert-butanol treatment 3 times. Later, the samples were dried with a critical point dryer (EM CPD300, Leica, Germany) and then stuck to a copper table. A sputter coater (SCD 500, Bal Tel, Germany) was used to plate a 10-nm thick gold film on the samples. These samples were then sent to the test center of Yangzhou University for scanning electron microscopy (XL-30, Philips, Holland) for analysis purposes.

### 2.5. Statistical Analysis

Data were analyzed using the mixed models procedure of Statistical Product and Service Solutions software (SPSS version 16.0; SPSS Inc., Chicago, IL, USA) using the following equation:
Y_ijk_ = µ + T_i_ + S_j_ + (T×S)_ij_ + R_k_ + e_ijk_(3)
where Y_ijk_ is the dependent variables, µ is the overall mean, T_i_ is the fixed effect of treatments, S_j_ is the fixed effect of sampling time, (T×S)_ij_ is the fixed effect of interaction between treatment and sampling time, R_k_ is the random effect of repetitions (k = 3 per treatment), and e_ijk_ is the random error term. Significance was declared at *p* < 0.05 and tendency at 0.05 < p < 0.1. Here, 24 h in vitro nutrient digestibility differences in each treatment were determined by Tukey’s multiple comparison test in a general linear model (GLM) procedure and were considered significant if *p* < 0.05. Test data were expressed as the mean and standard error of the mean (SEM).

## 3. Results

### 3.1. In Vitro Nutrient Digestibility of Different U. cannabina to L. chinensis Ratios at 24 h 

It can be seen from Table 2 that the different ratios of *U. cannabina* and *L. chinensis* had significant effects on 24 h in vitro dry matter digestibility (IVDMD) with *p* < 0.05. Among groups, group D and group E had lower IVDMD than the other three groups (*p* < 0.05). The 24 h in vitro digestibility of crude protein (IVCPD) was observed to be significantly higher in group A than in groups B, D, and E (*p* < 0.05). The significant effect was also detected at 24 h for in vitro digestibility of neutral detergent fiber (IVNDFD) (*p* < 0.05) with maximum digestibility for group C and with the minimum digestibility for group E. On the other hand, 24 h in vitro digestibility of acid detergent fiber (IVADFD) was not significantly different among any of the groups (*p* > 0.05).

### 3.2. In Vitro Protozoa Counts of Different U. cannabina to L. chinensis Ratios 

The total number of protozoa in each group fluctuated within the range of 150,000/mL to 280,000/mL, with an overall trend of inclining at first and then declining, as shown in Figure 1. In the 24 h culture, the highest trend was observed in group D, and the lowest in group C. 

The proportion of genus *Entodinium* in the total count ranged from 70% to 90% (Figure 2). Group A had a significantly higher proportion of genus *Entodinium* than the other four groups (*p* < 0.05), and group D had the lowest proportion at 24 h. The range of *Diplodinium* was between 1.5% and 22%, and no significant difference in the ratios was noticed at 24 h (*p* > 0.05). The ratio of genus *Isotricha* ranged between 3% and 8%, and also did not differ significantly among the treatments at 24 h (*p* > 0.05).

### 3.3. In Vitro MCP Content of Different U. cannabina to L. chinensis Ratios 

The sampling time and the ratio of *U. cannabina* to *L. chinensis* had significant effects on the MCP content (Table 3; *p* < 0.05). By looking at the comparisons between groups at 1, 3, and 6 h, it was observed that the MCP contents of group A and C were significantly higher (*p* < 0.05). These contents were highest in group C at 24 h (*p* < 0.05). In addition, group E fluctuated greatly with time. 

### 3.4. Dynamic Changes of Rumen Microorganisms Attached to U. cannabina and Ultrastructure of Fibers Under Scanning Electron Microscopy

Figure 3 shows in vitro ultrastructural changes and microbial attachment of *U. cannabina* under scanning electron microscopy for 1, 3, 6, and 24 h in groups B, C, D, and E. At 1 h, a large number of microorganisms were attached to the surface of *U. cannabina* in each group. Fungal hyphae (B-1, C-1, D-1), cocci, and irregular-shaped bacteria (C-1, E-1) were also observed. The phloem was slightly degraded in group C, however, there was no obvious degradation in the other groups. At 3 h and 6 h, the number of microbial attachments gradually increased and varied, which formed a large number of reticular mycelia (B-3, C-3, E-6) in the phloem, accompanied by cell wall degradation products (B-6, C-6) with obvious differences in the degradation of different tissue structures (D-6, E-6), in addition, the degradation of D-3, E-3 is not obvious. At 24 h, the number of microbial attachments remained high and a large number of microorganisms accumulated in the phloem of the basic tissues and vascular bundles, leading to the degradation of phloem. A large number of cocci were attached to the secondary xylem fibers, leading to the gradual exposure of the pore vessels (B-24, C-24). The primary xylem phloem in group C was largely degraded, and a large number of vascular bundle structures fell off after the surrounding phloem was degraded. However, the secondary xylem fibers and vessels remained intact (C-24). In groups D and E, the secondary xylem duct structure was not completely observed (D-24, E-24).

## 4. Discussions

### 4.1. U. cannabina to L. chinensis Ratios on Substrate Degradation Rates In Vitro 

The degree of nutrient metabolism in the substrate can reflect the rumen microbial activity, as well as the fermentation degree of the substrate [26]. Jin et al. [13] demonstrated that *U. cannabina* can be used as the replacement for the traditional feed mixture of forage in the feeding experiments. In the current in vitro trial, the CP contents increased with the high ratio of *U. cannabina* (Table 1), as it contains higher protein content compared to *L. chinensis* (165 and 87.3 g/kg DM). The degradation rate of CP was high in group A, which might be attributed to the higher rumen digestible protein of *L. chinensis*. Interestingly, the degradation rate of CP was also higher in group C, which is inconsistent with other studies conducted by Zhang et al. [10] and Jin et al. [13]. The reason might be the 1:1 ratio promoting ruminal microbial activity related to nutrient digestibility.

It has been stated that the nutritional availability of cellulose varies according to digestibility factors, which are related to the availability of lignin contents [27]. The lignin contents, along with indigestible carbohydrate fractions, have an opposite correlation with the non-fiber carbohydrates (NFCs) [28]. Meanwhile, rumen fungi preferentially adhere to thick-walled lignified tissue to help fiber degradation. Consequently, the presence of a large amount of cellulose in the substrate promotes its growth and fiber degradation [29]. In this experiment, the degradation rate of NDF was partially increased in group C due to the higher proportions of NDF and NFCs and lower contents of lignin (Table 1), which might have promoted the cellulose degradation by microorganisms [30].

### 4.2. Effects of U. cannabina to L. chinensis Ratios on MCP Content and Protozoa Counts In Vitro

Rumen protozoa is one of the main groups of rumen microbes in ruminants. Among these protozoa, ciliates consist of hundreds genera, of which *Entodinium*
*spp.* accounts for about 65% to 98% of the total ciliates that play certain roles in the degradation process of plant cell walls [31]. Wang and McAllister [32] reported that cellulolytic enzymes also exist in protozoa, which accelerate cell separation by lysing fibrous tissue and promoting cell wall rupture, followed by swallowing and digestion, as a single worm can swallow 148 ng of cellulose at a digestion rate of 320 pg/protozoa/min. The rumen protozoal population is manipulated by several factors, and the number of total protozoa varies from 10^5^/mL to 10^6^/mL in rumen liquid [33]. In the current experiment, the total number of protozoa in each group fluctuated between 15×10^4^/mL to 30×10^4^/mL, which falls under the allowed range. The total number of protozoa was lower in group A and group C when cultured in vitro for 24 h in the present trial. This may be due to the fact that *L. chinensis* as a culture substrate is more conducive to microbial growth. As time went on, bacterial proliferation reached a plateau, and protozoal engulfing of bacteria was also increased; later on, the total number of protozoa decreased due to the autolysis of protozoa. Jouany [34] reported that ciliate proteolytic enzymes are more vigorous than bacteria, with higher proteolysis capacity of insoluble proteins than soluble proteins [35]. Several factors, such as the double deamination activity, engulfment behavior, autolysis, and high deamination activity of protozoa, may aggravate the inefficient circulation of nitrogen in the rumen, leading to the reduced nitrogen utilization rate. Thus, it is wise to regulate protozoal biomass in order to increase MCP synthesis, which can be verified in the subsequent MCP contents. 

Additionally, the *Entodinium* was found to have a larger percentage (over 70%) as compared to other genera in the culture medium, with significant differences observed among different groups. The proportions of *Entodinium* and *Isotricha* were lowest in group C at 24 h, however an opposite trend was observed for *Diplodinium*, which showed increased NDF digestibility in group C at 24 h. These results are also inconsistent with Franzolin and Dehority [36] and Wang et al. [37]. It can be concluded that the ratio of *U. cannabina* to *L. chinensis* shifted the profile of the protozoa, community as well as the microbial population.

Rumen MCP contents are synthesized by rumen microbes through complex fermentation processes, and are the main nitrogen source for the host animal, providing about 40% to 80% of nitrogen requirements [38,39]. A variety of factors, such as the diet structure and composition, fermentation environment, and digesta flow rate, affect the amount of rumen microbial protein synthesis. The results of this experiment showed that the MCP contents of group A and group C were higher at 1, 3, 6, and 24 h. This may be due to the fact that *L. chinensis* is a high-quality forage grass, and as a substrate it is more conducive to the growth of rumen microorganisms. As the ratio of *U. cannabina* increased and *L. chinensis* inversely decreased in the substrate, the amount of MCP was also decreased directly. While in group C(50:50), the CP content of *U. cannabina* was higher than that of *L. chinensis*, the NDF content was lower than that of *L. chinensis* (Table 1), these results show that nutrients of *U. cannabina* and *L. chinensis* had certain complementary interaction effects [40], which led to an increased synthesis of rumen microbial proteins. These results also proved that the MCP content of this group was higher than that of group A through the culturing process. On the other hand, the MCP content in group E fluctuated greatly with the passage of culture time. The possible reason might be due to the certain pharmacological effects of *U. cannabina* on microorganisms [17].

### 4.3. Ultrastructural Variation of Fibers of U. cannabina in Different Ratio Groups

Liu et al. [41] reported that there was a positive correlation between fiber degradation and the number of attached microorganisms. Gonzalez-Munoz et al. [42] used scanning electron microscopy to observe the changes of Festulolium leaf tissue with time in vitro. They found that the degree of degradation was positively correlated with the passage of time, and the tissue composed of primary cell wall was basically degraded at 48 h. In this experiment, at 1 h it can be seen from the scanning electron microscopy that there was no obvious degradation mark in the *U. cannabina* of each group, because the tissue contains a surface layer that is resistant to attachment, and therefore resistant to degradation [43]. Finally, at 24 h, the tissue surface corrosion of each group was quite obvious. Compared with other groups, the whole vascular bundle structure of group C showed obvious shedding after degradation of the surrounding parenchyma. The phloem had been completely degraded and holes formed one by one at the degradation points. Akin and Amos [44] demonstrated that the mesophyll tissue and phloem fibers in the fescue were preferentially degraded compared to the peripheral vascular bundle sheath and epidermal tissue, leading to complete degradation of the epidermis and parenchyma. These results are in accordance with the *U. cannabina* degradation status in our experiment. Meanwhile, Jung et al. [45] also found that the epidermis, thick-horned tissue, green tissue, formation layer, and primary xylem thin-walled tissue of the alfalfa stalk can be rapidly degraded by rumen microorganisms, whereas the lignified tissue was only partially degraded after 96 h of fermentation; however, secondary xylem ducts were almost completely non-degradable. Among the treatments in the current study, group C was the most corroded and degraded by rumen microbes. This is because the rumen microbial fiber degradation efficiency was improved, which is consistent with the NDF digestibility results, as described above. As far as group D is concerned, it had the lowest degree of degradation, which might be due to the lower *L. chinensis* content in the substrate (Table 1) and the presence of stratum corneum in the stem, resulting in lower microbial degradation efficiency [46]. As for the slight degradation of the epidermis at 1 h in this experiment, it most likely was caused by the edge effect [47]. During the pulverization process, the epidermis was mechanically damaged and the microorganisms tended to adhere to the notched fragments [48].

The degradation of rumen roughage is mainly due to the fermentation of microorganisms, and the plant–microbe interactome is critical for the utilization of plant nutrients in rumen [49,50]. Ruminal fiber catabolic strains can degrade plant cells from the inside to the outside by using secreted enzymes. Cell walls with high degrees of lignification are difficult to degrade in the rumen; however, non-lignified parenchyma cells are easily degraded by microorganisms [51,52]. Cheng et al. [53] found that the adhesion of rumen bacteria to feed was dynamic; rumen fungi can not only grow rhizoids that penetrate the stratum corneum and lignin of plant cell walls, but also produce highly active cellulases that degrade plant cell walls [54,55,56]. 

Presently, the scanning electron micrograph of *U. cannabina* showed that there were fewer microorganisms attached at 1 h, and the corresponding degradation rate was relatively lower. In general, rumen bacteria can rapidly attach to plant fragments, which play a vital role in fiber degradation, especially those of the cocci and irregularly shaped bacteria that are closely attached to the forage [43]. For example, *Ruminococcus flavefaciens* (*R. flavefaciens*), *Ruminococcus albus* (*R. albus*), and *Fibrobacter succinogenes* (*F. succinogenes*) account for more than 70% of the cells that closely adhere to the bacteria [23,57]. By observing the morphology of attached microorganisms, it was speculated that cellulose-degrading bacteria might also be present, such as *R. flavefaciens*, *F. succinogenes*, and *Butyrivibrio fibrisolvens*. At 3 h, with the microbial proliferation, the number of attached microorganisms on the surface of *U. cannabina* increased significantly, accompanied by the degradation of plant fragments. At 6 h, extracellular complexes synthesized by some rumen microbial species, which are known as appendages or small cellular extensions in *R. albus**,* were observed and apparently attached bacterial cells to the substrate [58]. At 24 h, the number of microbial attachments in each group remained high. Gonzalez-Munoz et al. [41] reported that the development of the extracellular complex was higher at 24 h of *Festulolium* incubation, when the maximum xylanase activity occurred in our experiment.

## 5. Conclusions

The current experiment demonstrated that a ratio of 50:50 of *U. cannabina* to *L. chinensis* increased the ruminal MCP content, decreased the protozoa population, and shifted the profile of the protozoal community, resulting in enhanced growth of rumen microorganisms in vitro. Additionally, the NDF digestibility of the substrate and the corrosion of the fiber microstructure by rumen microbes in vitro was also improved with the ratio of 50:50. These results together consistently showed that the ratio of 50:50 of *U. cannabina* to *L. chinensis* is comparatively beneficial to the growth and proliferation of rumen microorganisms and the degradation rate of the substrate in vitro. This study discovers that *U. cannabina* could potentially be used as a substitute for *L. chinensis* to alleviate the shortage in quality forage in ruminant production, which needs further investigation through feeding trials with ruminant species, such as sheep or cows. 

## Figures and Tables

**Figure 1 animals-10-00335-f001:**
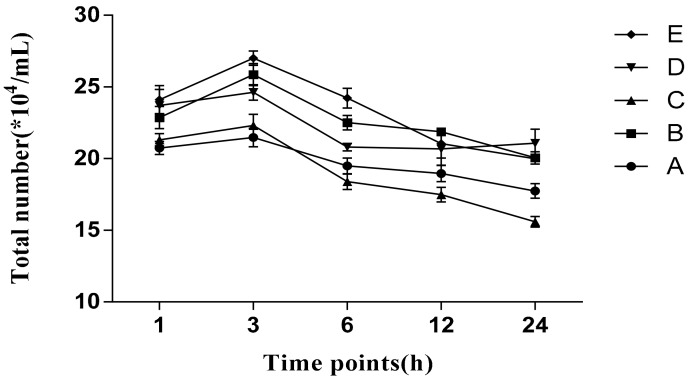
Variation of the total protozoal counts in culture medium. Ratio of *U. cannabina* to *L. chinensis* in group A (0:100), group B (30:70), group C (50:50), group D (70:30), and group E (100:0).

**Figure 2 animals-10-00335-f002:**
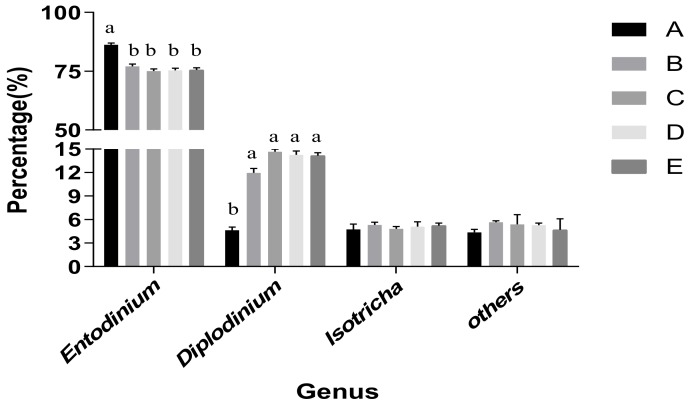
Profile of protozoal community in culture medium. Ratio of *U. cannabina* to *L. chinensis* in group A (0:100), group B (30:70), group C (50:50), group D (70:30), and group E (100:0). ^a-b^ Bar charts within same genus with different superscripts differ significantly at *p* ≤ 0.05.

**Figure 3 animals-10-00335-f003:**
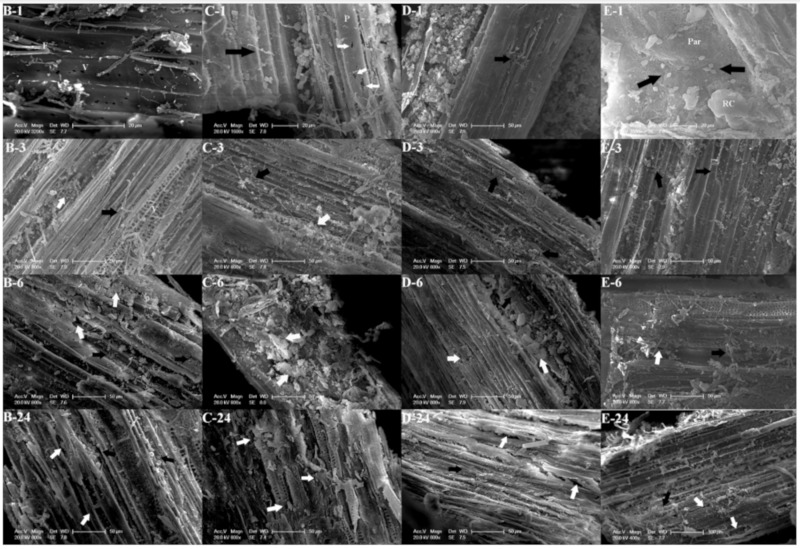
Electron scanning micrographs showing microbial attachment and cell wall degradation of *Urtica cannabina* in different groups at 1 h, 3 h, 6 h, and 24 h: results at 1 h (B-1, C-1, D-1, E-1); results at 3 h (B-3, C-3, D-3, E-3); results at 6 h (B-6, C-6, D-6, E-6); results at 24 h (B-24, C-24, D-24, E-24). F, fiber (B-1); V, vessel (B-1); P, phloem (C-1) Par, parenchyma (E-1); RC, ray cell (E-1). Black arrows, ruminal microorganisms; white arrows, cell wall degradation.

**Table 1 animals-10-00335-t001:** Ingredients, composition, and nutritive value of in vitro culture substrates.

Item	Group Treatment				
	A	B	C	D	E
Ingredient (%)					
*U. cannabina*	0	30	50	70	100
*L. chinensis*	100	70	50	30	0
Total	100	100	100	100	100
Nutritive level (g/kg)^1^					
CP	87.3	110.61	126.15	141.69	165
EE	25.4	26.48	27.2	27.92	29
NDF	670	583.6	526	468.4	382
ADF	302	301.7	301.5	301.3	301
NFC	153	177.6	194	210.4	235
Lignin	86.5	71.98	62.3	52.62	38.1
Ash	63.8	101.36	126.4	151.44	189
Ca	5.1	13.89	19.75	25.61	34.4
P	1.2	5.64	8.6	11.56	16

^1^ Nutritional value was measured; CP = crude protein; EE = ether extract; NDF = neutral detergent fiber; ADF = acid detergent fiber; NFC = non-fiber carbohydrates = 1000 - (CP + EE + NDF + ash); Ca = calcium; P = phosphorus. The five groups A–E have the following respective ratios of *U. cannabina* to *L. chinensis:* 0:100, 30:70, 50:50, 70:30, and 100:0, respectively.

**Table 2 animals-10-00335-t002:** Twenty-four hour in vitro digestibility of DM, CP, NDF, and ADF/% for the five groups A-E.

Items	Treatment Group					SEM	*P*-Value
	A	B	C	D	E		
IVDMD	68.81 ^a^	68.26 ^a^	68.01 ^a^	59.82 ^b^	58.40 ^b^	1.563	0.012
IVCPD	71.09^a^	65.27 ^b^	68.09 ^ab^	64.05 ^b^	64.94 ^b^	0.887	0.042
IVNDFD	66.20 ^ab^	66.53 ^ab^	67.28 ^a^	57.36 ^ab^	51.01 ^b^	4.062	0.031
IVADFD	27.63	29.73	31.47	26.84	29.80	2.442	0.436

DM, dry matter; IVDMD, in vitro digestibility of dry matter; IVCPD, in vitro digestibility of crude protein; IVNDFD, in vitro digestibility of neutral detergent fiber; IVADFD, in vitro digestibility of acid detergent fiber. Ratio of *U. cannabina* to *L. chinensis* in group A (0:100), group B (30:70), group C (50:50), group D (70:30), and group E (100:0); ^a,b^ values within a row with different superscripts differ significantly at *p* ≤ 0.05. SEM, standard error of the mean.

**Table 3 animals-10-00335-t003:** The effect of *U. cannabina* to *L. chinensis* ratios on the microbial crude protein (MCP) content in vitro (mg/mL).

Sampling Time/h	Treatment Group					SEM	*p*-Value		
	A	B	C	D	E		S	T	S×T
1	1.91 ^Aa^	1.13 ^Bb^	2.23 ^Aa^	0.68 ^Bc^	1.11 ^Ab^	0.137	<0.001	<0.001	<0.001
3	1.40 ^ABab^	1.49 ^ABab^	1.76 ^Aa^	0.78 ^Bc^	1.34 ^Ab^				
6	1.24 ^Ba^	1.68 ^Aa^	1.36 ^Ba^	1.16 ^Aa^	0.49 ^Bb^				
12	1.23 ^B^	1.32 ^AB^	1.3 ^B^	1.43 ^A^	1.57 ^A^				
24	1.57 ^ABab^	1.01 ^Bc^	1.82 ^Aa^	1.26 ^Aab^	1.41 ^Ab^				

*U. cannabina* to *L. chinensis* ratios in group A (0:100), group B (30:70), group C (50:50), group D (70:30), and group E (100:0). ^a-c^ Means within a row with different lowercase superscript letters differ at *p* ≤ 0.05; ^A, B^ means within a column with different uppercase superscript letters differ at *p* ≤ 0.05. SEM, standard error of the mean. S, sampling time; T, treatment group; S×T, interaction between sampling time and treatment group.

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
