# Peer review of "Effects of Urtica cannabina to Leymus chinensis Ratios on Ruminal Microorganisms and Fiber Degradation In Vitro"

_animals, 2020, doi:10.3390/ani10020335_

Round 1

Reviewer 1 Report

This is an interesting in vitro study about the inclusion of U. cannabina in rations for ruminants. However, I have detected some major concerns that may need improvement before considering the manuscript for publication; mainly, the sections Materials and methods and results need an improvement to make it easier to folow the developement of the trial and the presentation of the results. Also, some extensinve English editing is necessary, especially in materials and methods and results, and some part of the discussion.

L7-11: E-mails of coauthors are missing

Abstract: limit is 200 words, please shorten.

L42: group C is mentioned twice

L48: in the keywords, please write the full scientific name of the plants.

Introduction: some english editing is needed

L80: please, take into account the archaea and virus when accounting rumen microorganisms

L87: what is the improper ratio?

Materials and methods: as stated before, a general improvement is needed in this section in order to better follow the design of the experiment

L122-130: specify the number of bottles used in the experiment. There was a culture medium preparation or the artificial saliva (please, refer to it as artificial saliva and not artificial saliva salt) was mixed with the rumen fluid? The procedure needs to be clarified. How the flask was charged with CO2? How the CO2 was introduced in the medium?

L131-135: More details are needed in order to understand what was done.

L150: Please, cite the reference in the format of the journal

L157: please define MFS

L162: what do you mean by "several protozoa"?

L165: even if the reference is cited, a line explainng the method would help understanding how the microbial protein was determined

L175: The replicate is the random effect? That does not make sense, what do you mean by "replicate"?

L195: please, add the units to the number of protozoa

Results: please check the number of sections, Results shoul be number 3. Also, titles of subheading do not clarify what you are talking about. Results should be explained more in detail. In tables, SEM shoul be expressed with one decimal more than the means. In tabe 4, a P value is stated for each time point and then several P values are expressed for the whole. Please, clarify to what they refer in the Table and in the statistical analysis section.

L225-233: There are 16 pictures in the Figure 3, more explanaitions should be included in this section or the number of pictures should be reduce in order to improve the understanding of the section

Discussion: The same order as in the presentation of the results must be maintained (degradation, protozoa and MCP, ultrastructure of fiber). I think along the section bacteria and microorganism are used indistinctively; please refer to bacteria when only this group is involved as protozoa are also microorganisms.

Also, check the numbering of the sections, discussion should be the 4.

L288-291: Please, check the spelling of the bacterial genus and species

L301: Are you sure that there are hundreds of protozoa genera in the rumen?

Author Response

Review 1.

L7-11: E-mails of coauthors are missing

A: It has been added.Please see L8-13.

Abstract: limit is 200 words, please shorten.

A: Abstract meets the current words limit.Please see L24-36.

L42: group C is mentioned twice

A: It has been corrected.

L48: in the keywords, please write the full scientific name of the plants.

A: have been revised. Please see L37.

Introduction: some English editing is needed

A: English editing has been made.

L80: please, take into account the archaea and virus when accounting rumen microorganisms

A: thanks for the suggestion, we have mentioned in the manuscript.Please see L70.

L87: what is the improper ratio?

A: This means that the ratios are comparatively lower than the optimum ratio in the fermentation and digestibility. 

Materials and methods: as stated before, a general improvement is needed in this section in order to better follow the design of the experiment

A: already revised as the suggestion.

L122-130: specify the number of bottles used in the experiment. There was a culture medium preparation or the artificial saliva (please, refer to it as artificial saliva and not artificial saliva salt) was mixed with the rumen fluid? The procedure needs to be clarified. How the flask was charged with CO2? How the CO2 was introduced in the medium?

A: the detailed has been explained in the manuscript. Please see L118-132.

L131-135: More details are needed in order to understand what was done.

A: We have revised and provided more details. Please see L142-146 and L154-159.

L150: Please, cite the reference in the format of the journal

A: already revised as the suggestion through the text.Please see reference 21.

L157: please define MFS

A: M.F.S. is M (methylgreen), F (formaldehyde) and S (staining solution). The full name was added in the text. Please see L174-175.

L162: what do you mean by "several protozoa"?

A: There was a mistake here, already revised in the manuscript. Please see L185-187.

L165: even if the reference is cited, a line explaining the method would help understanding how the microbial protein was determined

A: We have explained that Trichloroacetic acid (TCA) precipitation method was used to detect the MCP (microbial protein) content in the manuscript. Please see L189-190.

L175: The replicate is the random effect? That does not make sense, what do you mean by "replicate"?

A: the description is not clear here. We have already revised. Please see L200-209.

L195: please, add the units to the number of protozoa

A: The unit has been added. Please see L230-231.

Results: please check the number of sections, Results should be number 3. Also, titles of subheading do not clarify what you are talking about. Results should be explained more in detail. In tables, SEM should be expressed with one decimal more than the means. In table 4, a P value is stated for each time point and then several P values are expressed for the whole. Please, clarify to what they refer in the Table and in the statistical analysis section.

A: We have changed the number of Results section to 3 and revised the titles of subheading.

Additionally, more details have been added in Results section. 

SEM has also been revised in the manuscript. And the Table 4 has been changed to Table 3,which has been made clearer.

L225-233: There are 16 pictures in the Figure 3, more explanations should be included in this section or the number of pictures should be reduce in order to improve the understanding of the section

A: more explanations have been included in this section. Please see L263-278.

Discussion: The same order as in the presentation of the results must be maintained (degradation, protozoa and MCP, ultrastructure of fiber). I think along the section bacteria and microorganism are used indistinctively; please refer to bacteria when only this group is involved as protozoa are also microorganisms. Also, check the numbering of the sections, discussion should be the 4.

A: Discussion section have mentioned in the order as in the presentation of the results.

L288-291: Please, check the spelling of the bacterial genus and species

A: already revised. Please see L390-394.

L301: Are you sure that there are hundreds of protozoa genera in the rumen?

A: yes, it is true. There are hundreds of protozoa genera in the rumen (Williams and Coleman, 1992), it is however that, not all the genera can be detected by morphology. There are only 6 protozoa genera, Entodinium, Diplodinium, Isotrich, Epidinium,Ophryoscolecinae and Dasytricharuminantium,which can be detected in this work.

Ref:

 Williams, A.G. and Coleman, G.S.Identification and Classification of Entodiniomorphid Protozoa,in The Rumen Protozoa; Springer: New York, NY, USA,1992;pp.4-85,ISBN:978-1-4612-7664-7.

Reviewer 2 Report

Many things had improved from the first submission. However after reading the content, this reviewer have to suggest whether authors could improve the title of the manuscript. This title is a little bit confusing for reader what would be the major content and focus. Although authors insert "microstructure" , the result about microstructure is believed not remarkable enough to stand forward. In addition, change in microstructure is not a fundamental proof of finding and supporting the resulted appropriate ratio.

Author Response

Dear reviewer

Thanks very much for your good suggestion.After co-authors' discussion,we have modified the title to 'Effects of Urtica cannabina to Leymus chinensis ratios on the ruminal microorganism and fiber degradation in vitro'.we think this title will let readers know what the mian content and focus of this research. 

Please check.

Kind regrads

Zhenbin Zhang

This manuscript is a resubmission of an earlier submission. The following is a list of the peer review reports and author responses from that submission.

Round 1

Reviewer 1 Report

This is an interesting in vitro study about the inclusion of U. cannabina in rations for ruminants. However, I have detected some major concerns that may need improvement before considering the manuscript for publication; mainly, the sections Materials and methods and results need an improvement to make it easier to folow the developement of the trial and the presentation of the results. Also, some extensinve English editing is necessary, especially in materials and methods and results, and some part of the discussion.

L7-11: E-mails of coauthors are missing

Abstract: limit is 200 words, please shorten.

L42: group C is mentioned twice

L48: in the keywords, please write the full scientific name of the plants.

Introduction: some english editing is needed

L80: please, take into account the archaea and virus when accounting rumen microorganisms

L87: what is the improper ratio?

Materials and methods: as stated before, a general improvement is needed in this section in order to better follow the design of the experiment

L122-130: specify the number of bottles used in the experiment. There was a culture medium preparation or the artificial saliva (please, refer to it as artificial saliva and not artificial saliva salt) was mixed with the rumen fluid? The procedure needs to be clarified. How the flask was charged with CO2? How the CO2 was introduced in the medium?

L131-135: More details are needed in order to understand what was done.

L150: Please, cite the reference in the format of the journal

L157: please define MFS

L162: what do you mean by "several protozoa"?

L165: even if the reference is cited, a line explainng the method would help understanding how the microbial protein was determined

L175: The replicate is the random effect? That does not make sense, what do you mean by "replicate"?

L195: please, add the units to the number of protozoa

Results: please check the number of sections, Results shoul be number 3. Also, titles of subheading do not clarify what you are talking about. Results should be explained more in detail. In tables, SEM shoul be expressed with one decimal more than the means. In tabe 4, a P value is stated for each time point and then several P values are expressed for the whole. Please, clarify to what they refer in the Table and in the statistical analysis section.

L225-233: There are 16 pictures in the Figure 3, more explanaitions should be included in this section or the number of pictures should be reduce in order to improve the understanding of the section

Discussion: The same order as in the presentation of the results must be maintained (degradation, protozoa and MCP, ultrastructure of fiber). I think along the section bacteria and microorganism are used indistinctively; please refer to bacteria when only this group is involved as protozoa are also microorganisms.

Also, check the numbering of the sections, discussion should be the 4.

L288-291: Please, check the spelling of the bacterial genus and species

L301: Are you sure that there are hundreds of protozoa genera in the rumen?

Author Response

Dear reviewer

Please see attachments

thanks!

Reviewer 2 Report

Please double check material and method section. There would be mistakes on titling and experimental materials for in vitro fermentation.

Authors may need to be more specific when to illustrate the rationale of the experimental objective. It is logically too early if author say the appropriate substitution ratio could be a solution for the supply shortage of good quality forage.

I think the particle size of forage for in vitro fermentation would be too fine if the forage were ground and sieved with less than 1 mm. In addition, rumen fluid used was the composite fluid from three goats? Need to justify why the goat were fed that basal diet since the basal diet affect the in vitro characteristic. 

Current experimental treatments(A~E) would not be appropriate (too wide) for finding practically appropriate substitution level. Do authors sure think that 50:50 ratio is the appropriate ratio? You need to justify more specifically.  

Author Response

Dear reviewer

thanks!
